# National Burden and Trends for 29 Groups of Cancer in Mexico from 1990 to 2019: A Secondary Analysis of the Global Burden of Disease Study 2019

**DOI:** 10.3390/cancers16010149

**Published:** 2023-12-28

**Authors:** Saul A. Beltran-Ontiveros, Jose A. Contreras-Gutierrez, Erik Lizarraga-Verdugo, Erick P. Gutierrez-Grijalva, Kenia Lopez-Lopez, Emilio H. Lora-Fierro, Miguel A. Trujillo-Rojas, Jose M. Moreno-Ortiz, Diana L. Cardoso-Angulo, Emir Leal-Leon, Jose R. Zatarain-Lopez, Hector M. Cuen-Diaz, Marisol Montoya-Moreno, Brisceyda Arce-Bojorquez, Juan L. Rochin-Teran, Daniel E. Cuen-Lazcano, Victor A. Contreras-Rodriguez, Ricardo Lascurain, Liliana Carmona-Aparicio, Elvia Coballase-Urrutia, Francisco Gallardo-Vera, Daniel Diaz

**Affiliations:** 1Centro de Investigación y Docencia en Salud, Universidad Autónoma de Sinaloa, Culiacán Rosales 80030, Sinaloa, Mexico; saul.beltran@uas.edu.mx (S.A.B.-O.); eriklizarraga@uas.edu.mx (E.L.-V.); diana.cardoso@uas.edu.mx (D.L.C.-A.); marisol.montoya@uas.edu.mx (M.M.-M.); brisceyda.arce@uas.edu.mx (B.A.-B.);; 2Cátedras CONACYT, Centro de Investigación en Alimentación y Desarrollo, A.C., Culiacán Rosales 80110, Sinaloa, Mexico; erick.gutierrez@ciad.mx; 3Laboratorio de Biomedicina Molecular, Facultad de Ciencias Químico Biológicas, Universidad Autónoma de Sinaloa, Culiacán Rosales 80019, Sinaloa, Mexico; kenia.lopez@uas.edu.mx; 4Instituto de Genética Humana “Dr. Enrique Corona Rivera”, Centro Universitario de Ciencias de la Salud, Universidad de Guadalajara, Guadalajara 44340, Jalisco, Mexico; migueltrojas@alumnos.udg.mx (M.A.T.-R.); miguel.moreno@academicos.udg.mx (J.M.M.-O.); 5Laboratorio de Genética y Biología Molecular, Facultad de Ciencias Químico Biológicas, Universidad Autónoma de Sinaloa, Culiacán Rosales 80019, Sinaloa, Mexico; emir.leal@uas.edu.mx; 6Unidad Académica de Criminalística, Criminología y Ciencias Forenses, Universidad Autónoma de Sinaloa, Culiacán Rosales 80040, Sinaloa, Mexico; victorcontreras@uas.edu.mx; 7Unidad de Vinculación Científica, Facultad de Medicina, Universidad Nacional Autónoma de México en el Instituto Nacional de Medicina Genómica, Tlalpan 14610, Ciudad de México, Mexico; rlascurain@facmed.unam.mx; 8Laboratorio de Neurociencias II, Instituto Nacional de Pediatría, Coyoacán 04530, Ciudad de México, Mexico; c_apariccio@ciencias.unam.mx (L.C.-A.); ecoballaseu@pediatria.gob.mx (E.C.-U.); 9Laboratorio de Biología Molecular y Bioseguridad Nivel III, Centro Médico Naval, Coyoacán 04470, Ciudad de México, Mexico; jfgallardo@ciencias.unam.mx; 10Facultad de Ciencias, Universidad Nacional Autónoma de México, Coyoacán 04510, Ciudad de México, Mexico

**Keywords:** burden of disease, cancer epidemiology, cancer mortality, malignant neoplasm, public health

## Abstract

**Simple Summary:**

Cancer is a significant contributor to morbidity and mortality worldwide. The purpose of this study was to analyze the cancer burden and trends of 29 groups of malignant neoplasms in Mexico by sex and age from 1990 to 2019. In 2019, there were 222.1 thousand incident cases and 105.6 thousand deaths due to cancer in the general population. The number of new cases and deaths from the 29 cancer groups increased between 10% and 436% from 1990 to 2019, with different patterns by sex and age. Breast, cervical, and colorectal cancers were the leading causes of death among women, while prostate, lung, and colorectal cancers were the leading causes of death among men. In Mexico, malignant neoplasms were the third leading cause of death in 2019, causing significant health loss. The existence of gender disparities emphasizes the need for cancer-specific targeted prevention, diagnosis, and treatment.

**Abstract:**

The global burden of cancer is on the rise, with varying national patterns. To gain a better understanding and control of cancer, it is essential to provide national estimates. Therefore, we present a comparative description of cancer incidence and mortality rates in Mexico from 1990 to 2019, by age and sex for 29 different cancer groups. Based on public data from the Global Burden of Disease Study 2019, we evaluated the national burden of cancer by analyzing counts and crude and age-standardized rates per 100,000 people with 95% uncertainty intervals for 2019 and trends using the annual percentage change from 1990 to 2019. In 2019, cancer resulted in 222,060 incident cases and 105,591 deaths. In 2019, the highest incidence of cancer was observed in non-melanoma skin cancer, prostate cancer, and breast cancer. Additionally, 53% of deaths were attributed to six cancer groups (lung, colorectal, stomach, prostate, breast, and pancreatic). From 1990 to 2019, there was an increasing trend in incidence and mortality rates, which varied by 10–436% among cancer groups. Furthermore, there were cancer-specific sex differences in crude and age-standardized rates. The results show an increase in the national cancer burden with sex-specific patterns of change. These findings can guide national efforts to reduce health loss due to cancer.

## 1. Introduction

Cancer is a significant cause of morbidity and mortality worldwide [1]. In 2019, it was the second leading cause of death, with an estimated 23.6 million new cases (17.2 million when excluding non-melanoma skin cancer) and 10.0 million deaths [2]. The incidence and mortality rates of cancer have been increasing, with an expected rise to 28.4 million cases by 2040, a 47% increase from 2020 [3]. If exposure to behavioral and environmental risk factors contributing to the cancer burden continues to increase, the global burden of cancer may worsen [4].

The United Nations and the World Health Organization have recognized the need to reduce the burden of cancer and develop strategies for national-level cancer control planning and implementation through the Sustainable Development Goals. However, there are regional and national differences in cancer morbidity and mortality associated with varying levels of exposure to population risk factors due to socioeconomic changes [5]. Therefore, understanding the local cancer epidemiology is crucial for informing cancer control efforts, including prioritizing resource allocation, implementing public health policies, and improving health system planning [6]. Previous studies in Mexico have described the specific burden of certain cancer groups at both national and state levels [7,8,9,10,11,12]. Furthermore, various studies have evaluated the national burden of cancer, encompassing multiple groups of malignant neoplasms from 1970 to 2015 [13,14,15,16]. However, due to the changing pattern and increasing trend of cancer burden, it is crucial to provide timely, reliable, and accurate estimates of the national burden of cancer.

The objective of this study is to compare the incidence and mortality rates of 29 types of cancer in Mexico in 2019 by sex and age, with trends from 1990 to 2019. The data used in this study were from the Global Burden of Disease Study 2019 (GBD 2019), which is a comprehensive global effort to systematically assess the major causes of health loss at the global, regional, and national levels [17]. The presented results provide updated information for a better understanding of the current context of evolution and trends of the national burden of cancer in Mexico. This information can be used to implement concrete public health actions to promote cancer control, diagnosis, and treatment programs.

## 2. Methods

### 2.1. GBD Study Overview

This observational study is a secondary analysis of results published in the Global Burden of Disease Study 2019 (GBD 2019). The GBD 2019 is produced by the Institute of Health Metrics and Evaluation (IHME), which includes a vast network of global collaborators, institutions, and partnerships involved in health policy and practice. This annual iteration was published in a series of four Capstone Papers [17,18,19,20], which describe in detail the trends in fertility, demography, and health loss due to 369 diseases and injuries and their corresponding 87 risk factors from 1990 to 2019. The estimates produced by the GBD 2019 represent the most comprehensive assessment of global health, providing timely and reliable results for 204 countries and territories by sex and age group. This study was produced and supervised by members of the GBD Collaborative Network from Mexico and adheres to GBD protocol.

### 2.2. Estimation of the Burden of Cancer in Mexico and Reporting Standards

The GBD 2019 report incorporates both fatal and non-fatal outcomes and provides a comprehensive hierarchical list of causes with four levels. Among the 22 groups at Level 2, ‘neoplasms’ (both benign and malignant) are included. At Level 3, there are 29 malignant neoplasms (cancers) and one group of benign neoplasms. The source for this information is https://vizhub.healthdata.org/gbd-compare/, accessed on 31 October 2023.

A detailed description of the approach, modeling framework, and steps taken by the GBD to generate the specific burden for neoplasms (total, benign, and malignant) is provided elsewhere [17]. Previous GBD studies have reported the methodology used to estimate the global, national, and regional burden of the 29 cancer groups, including case definition, data input sources, data processing, and statistical analysis [2,6]. The specific analytical flowchart used to produce cancer estimates is available online at https://ghdx.healthdata.org/gbd-2019/code/cod-2 (accessed on 2 November 2023).

This secondary analysis utilized publicly available data from the Global Health Data Exchange (GHDx), an online repository of the Institute for Health Metrics and Evaluation (IHME) (https://ghdx.healthdata.org, accessed on 11 August 2023). The GHDx provides GBD results for use in scientific publications, health policy, dissemination, and reporting. The results for Mexico were downloaded as CSV files from the GHDx online query tool (https://vizhub.healthdata.org/gbd-results/, accessed on 11 August 2023). To depict the cancer burden in Mexico between 1990 and 2019, we utilized crude counts of incidence and deaths caused by 29 groups of malignant neoplasms. Furthermore, we obtained crude and age-standardized rates per 100,000 individuals at the national level for each year, sex, and 5-year age groups. To compare the trends in these measures, we also downloaded the annual percentage change from 1990 to 2019.

Tables and figures were utilized to provide a comparative description of the burden of 29 cancer groups in Mexico for 2019, along with trends from 1990 to 2019. The data are presented as point estimates with 95% uncertainty intervals (95% UI), which were generated by the GBD 2019.

## 3. Results

### 3.1. National Burden of Disease Due to 29 Malignant Neoplasms during 2019

In Mexico, the 29 groups of malignant neoplasms mapped by the GBD caused an estimated 222,060 new cases in 2019. According to their national incidence (thousand new cases, 95% CI), non-melanoma skin cancer (41.3, 34.3 to 48.7), prostate cancer (27.1, 20.6 to 36.0), breast cancer (24.4, 19.9 to 29.9), colorectal cancer (17.5, 15.0 to 20.1), cervical cancer (12.2, 9.6 to 16.5), and stomach cancer (11.3, 9.8 to 13.0) were the top six ranked cancer groups in 2019 and contributed 60.2% of the total cases due to malignant neoplasm. In contrast, with incidence estimates ranging from 213 to 439 new cases, mesothelioma, other pharynx cancer, and nasopharyngeal cancer had the lowest incidence in Mexico during 2019 (Figure 1a and Table 1). The age-standardized incidence rate of these 29 malignant neoplasms varied substantially between 0.18 and 35.66 cases per 100,000 people. According to Appendix A, non-melanoma skin cancer and nasopharyngeal cancer had the highest and lowest rates, respectively.

In 2019, there were an estimated 105,591 deaths attributed to 29 types of cancer in Mexico. Of these malignant neoplasms, 52.9% were caused by six types of cancer: lung (11,002 deaths, 10.4% of the total), colorectal (10,518, 9.9%), stomach (10,095, 9.6%), prostate (9257, 8.8%), breast (8097, 7.7%), and pancreatic (6853, 6.5%). In 2019, the fewest cancer deaths were attributed to mesothelioma, other pharynx cancer, and nasopharyngeal cancer (range: 174–353 deaths; see Table 1 and Figure 1c). The highest age-standardized death rates per 100,000 people were observed for lung, colorectal, stomach, and prostate cancer (range: 8.78 to 9.74; see Appendix A).

### 3.2. National Trends in Crude Incidence and Mortality Due to 29 Malignant Neoplasms from 1990 to 2019

Figure 1a shows that seven cancer groups maintained their incidence ranking at the national level between 1990 and 2019, including non-melanoma skin cancer, breast cancer, stomach cancer, and pancreatic cancer in the top 10. Furthermore, among the 13 cancer groups that increased their ranking, prostate, colorectal, uterine, and thyroid cancer showed the most significant advancement. In contrast, the greatest declines were observed in cases of laryngeal cancer and Hodgkin lymphoma, gallbladder cancer, and brain cancer. Despite the different patterns of change observed, all 29 groups of cancer showed a positive percentage of annual change from 1990 to 2019, albeit variable (Figure 1b). The incidence increases ranged from 27% (−1 to 97) for cervical cancer to 436% (264 to 541) for melanoma (Appendix A). Several cancer types experienced significant increases, including colorectal, prostate, uterine, thyroid, breast, testicular, and liver cancer, with increases ranging from 310% to 421%.

The national ranking of cancer-related deaths revealed that six cancer types (lung, esophageal, lip, oral cavity, uterine, testicular, and nasopharynx) maintained their positions from 1990 to 2019, while 13 cancer types moved up in rank (Figure 1c). Colorectal cancer experienced the highest increase, moving from the seventh to the second position. However, 10 cancer groups experienced a decrease in rank, with Hodgkin lymphoma, cervical cancer, leukemia, and brain cancer showing the largest decreases. It is worth noting that all 29 cancer groups had a positive percentage annual change in mortality from 1990 to 2019 (refer to Figure 1d). Liver cancer, colorectal cancer, melanoma, myeloma, kidney cancer, and ovarian cancer had the six highest increases in the numbers of deaths they caused, with values ranging from 232% to 314% (Appendix A).

### 3.3. National Cancer-Specific Trends from 1990 to 2019 in Age-Standardized Incidence and Mortality Rates

From 1990 to 2019, the age-standardized incidence and mortality rates (per 100,000 people) of 29 cancer groups in Mexico showed a contrasting pattern of change. Appendix A shows that 17 out of 29 malignant neoplasms had an increasing trend in age-standardized incidence rate during this period. Testicular cancer (190.7%), melanoma (116.8%), and colorectal cancer (89.2%) showed the largest increases, as shown in Appendix A. In contrast, the age-standardized incidence rates per 100,000 for cervical cancer decreased by 47.9%, gallbladder cancer by 42.7%, laryngeal cancer by 37.6%, and lung cancer by 32.8%.

Over the period, 16 out of 29 cancer groups had a decrease in their age-standardized death rates, but with wide variability in their time series (Figure 2). According to Appendix A, there were significant decreases in the incidence of cervical cancer, laryngeal cancer, gallbladder cancer, Hodgkin lymphoma, stomach cancer, and lung cancer (ranging from −35.5% to −58.3%). However, liver cancer (52.1%), melanoma (44.3%), and colorectal cancer (41.7%) showed substantial increases. To aid in interpreting the intricate patterns of results, Appendix A summarizes the percentage of change in age-standardized rates, ordered from the cancer with the largest increase to the cancer with the largest decrease. The figure illustrates that the percentage change varied by measure and cancer group.

### 3.4. National Burden of 29 Cancer Groups by Sex and Age Group in 2109

In 2019, the burden of disease due to cancer in Mexico exhibited a distinct age-specific distribution by sex. The crude incidence was lower in the early age groups (1–24 years) for both sexes. However, in women, there was a rapid increase in cancer incidence starting at 35–39 years of age. The highest number of cases was concentrated between 40 and 84 years of age, with a peak of 14,853 new cases in the 60–64 age group. In contrast, the number of incident cases among men increased gradually and occurred at older ages, starting at 45–49 years and peaking at 70–74 years with 12,291 new cases (Figure 3a). A similar pattern was found for mortality, with the highest number of deaths occurring in the 70–74 age group for women (5999 deaths) and in the 75–79 age group for men (7146 deaths) due to cancer (Figure 3b).

To analyze sex differences among the 29 cancer groups, we compared the crude counts and age-standardized rates per 100,000 people (Table 1 and Appendix A). Breast cancer (24,312 cases), non-melanoma (21,263 cases), and cervical cancer (12,195 cases) were the most common types of cancer in women, while prostate cancer, melanoma, and colorectal cancer (with a range of 9426 to 27,097 cases, Appendix A) were the most common in men. Prostate cancer was the leading cause of incident cases in men, with an age-standardized rate of 52.3 (40.0 to 70.1) per 100,000 people. In women, breast cancer was the leading cause with an estimated rate of 36.81 (30.02 to 45.01) cases per 100,000 people, while men had a negligible rate of 0.23 (0.19 to 0.29 cases) (Appendix A).

Breast cancer caused an estimated 8024 deaths among women in Mexico in 2019, followed by cervical cancer and colorectal cancer in second and third place with 6104 and 4917 deaths, respectively. Among men, prostate cancer was the leading cause of death from malignant neoplasms with 9256 deaths, followed by lung and colorectal cancers with 7176 and 5601 deaths, respectively (refer to Figure 3c). The age-standardized death rate varied significantly by cancer group in both sexes. Prostate cancer had the highest death rate among men, with 19.4 (14.8 to 26.7) deaths per 100,000 people. Among women, breast cancer had the highest death rate, with 12.5 (10.3 to 15.2) deaths per 100,000 people. The death rates for other neoplasms varied from 0.07 to 9.53 and 0.1 to 13.8 deaths per 100,000 people in women and men, respectively (Figure 3d).

Finally, based on the incidence (Appendix A) and mortality (Figure 4) rates per 100,000 people, there was a contrasting pattern of age-specific burden between sexes that varied by cancer type in Mexico in 2019. Overall, except for brain cancer, leukemia, Hodgkin lymphoma, and testicular cancer, most malignant neoplasms tended to have the highest burden in older ages. Several cancer groups, including bladder, brain, esophageal, Hodgkin lymphoma, kidney, larynx, lip and oral cavity, mesothelioma, nasopharynx, other pharynx, and lung cancer, had higher rates in men than in women.

## 4. Discussion

Although neoplasms (benign and malignant) moved from the second to the third leading cause of death in Mexico between 1990 and 2019, mortality rates caused by this group of diseases increased by 126.05% during this period. This increase was driven by the 29 groups of malignant neoplasms reported in this study (source: https://vizhub.healthdata.org/gbd-compare/, accessed on 10 August 2023). Our study found that lung cancer remained the leading cause of mortality from 1990 to 2019. Colorectal cancer moved up from seventh to second place, followed by stomach cancer, which dropped from second to third. Prostate cancer, breast cancer, and pancreatic cancer completed the list of the six most common causes of cancer death in Mexico in 2019. This pattern differs partially from a previous study that identified lung cancer, stomach cancer, liver cancer, prostate cancer, breast cancer, and cervical cancer as the top six causes of cancer-related deaths between 2000 and 2013 [15]. These findings may indicate a shift in the mortality pattern of cancer over the past two decades.

Other studies based on national databases from Mexico [7] have confirmed the rise in colorectal cancer mortality, which has also been documented globally [21]. The rise in colorectal cancer mortality is linked to a trend of higher incidence, even in early onset cases (<50 years of age) [22]. However, the absence of national screening strategies in Mexico may worsen this trend due to the detection of a large number of new cases at advanced stages, leading to higher mortality rates. Our results indicate that in 2019, the top six ranked cancer groups in Mexico were non-melanoma skin cancer, prostate cancer, breast cancer, colorectal cancer, cervical cancer, and stomach cancer among the 29 groups of malignant neoplasms in terms of incidence. The rise in cancer morbidity and mortality may be linked to various behavioral and environmental risk factors, including smoking, alcohol consumption, diet, radiation exposure, certain infections, and hormonal imbalances [23]. In Mexico, the prevalence of risk factors such as alcohol and tobacco consumption is high [15]. Therefore, further studies are necessary to determine the association between cancer mortality and risk factors.

Cancer affecting non-reproductive tissues has a higher incidence and mortality rate in males, resulting in roughly double the mortality rate compared to females [24]. Previous studies have shown that female cancer patients tend to have better survival rates than males [25]. However, in the Mexican population, overall mortality was slightly higher in females. The incidence of cancer in males is understudied despite the well-known disparity. This may be due to researchers assuming that known causes explain the disparity. According to GBD 2017 estimates, non-melanoma skin cancer was the most prevalent form of cancer among women globally, followed by breast cancer. In males, non-melanoma skin cancer had the most significant impact, followed by tracheal, bronchus, lung, and prostate cancer [6]. Our results showed that in comparison to this global trend, breast cancer has become the most prevalent cancer among females in Mexico, surpassing non-melanoma skin cancer. Cervical cancer, colorectal cancer, and ovarian cancer followed in frequency. Among males, prostate cancer has overtaken non-melanoma skin cancer, followed by colorectal, tracheal, bronchus, and lung cancer. The reasons for this differential pattern in Mexico remain unclear. However, the reasons for the differences in cancer rates and outcomes between sexes are not fully understood. Biological sex significantly influences organismal development and physiology, affecting processes such as cell signaling, metabolism, and immune responses [26,27]. The higher incidence of gallstones among women is likely the reason for the excess risk of gallbladder and biliary tract cancer in females [28]. Similarly, factors such as smoking and occupational exposures can be attributed to the higher portion of the male excess in urinary bladder cancer [29]. Additionally, the incidence of thyroid cases is more than double in women than that in men, a disparity that has been documented in other countries. Although the biological causes of this difference are not yet conclusive, it is speculated that non-biological factors may play a role. For instance, there is a possibility that women have more opportunities for incidental detection in clinical settings [30].

As anticipated, both sexes exhibited a higher incidence and mortality at older ages. Aging has been shown to accelerate cancer mortality, which may be influenced by various factors such as comorbidities, less intense detection, and lower likelihood of undergoing aggressive treatment [31]. In recent decades, Latin America has experienced a significant increase in life expectancy, resulting in a growing population of older adults. The rise in incidence and mortality rates in Mexico since 1990 can be partially attributed to this aging population. As a result, more individuals are expected to be diagnosed with cancer and experience cancer-related fatalities. A previous study demonstrated that advanced age may influence oncologists to avoid intensive cancer therapy, even in cases where patients are highly functional and have no comorbidities [32]. As a result, this group may have a worse prognosis and be undertreated, contributing to the mortality rate. Although our study did not categorize age groups, our results are consistent with global trends. The age-specific distribution showed that cancer incidence among children and teenagers was notably lower compared to older populations [33]. Furthermore, certain types of cancer disproportionately affect specific age groups. Leukemia is the primary contributor to cancer incidence among young individuals in the child and teenage age range [34]. A previous study in Mexico found that leukemia and other malignant neoplasms accounted for almost 70% of the cancer burden in younger age groups [14]. The present study shows a significant change in the prevalence of leukemia over time, with a decline in its ranking from fifth in 1990 to ninth in 2019. The changing landscape of cancer epidemiology suggests possible alterations in risk factors, diagnostic capabilities, or treatment modalities over time [35].

Lung cancer is the primary cause of death among respiratory system cancers, but its incidence in Mexico has decreased since 1990. Tobacco smoke is a significant contributing factor in the development of lung cancer, and it is widely recognized. Approximately 90% of lung cancer cases in men and 78% in women are estimated to be caused by tobacco smoke [36]. Tobacco consumption in Mexico has declined over the past few decades, especially among males [8]. Mexican healthcare systems aim to effectively address this issue by implementing advertising campaigns and initiatives to raise public awareness, particularly regarding lung cancer.

Colorectal cancer has a higher incidence among gastrointestinal tract cancers, and this incidence has increased nationally in recent years. This trend is consistent with the global trend, as the global incidence of colorectal cancer has more than doubled from 1990 to 2019 [21]. Other low- and middle-income countries, like Mexico, are also experiencing an increase in incident cases of colorectal cancer. In developed countries, there has been a trend of either decrease or stabilization [37]. It is possible that these trends are influenced by an increase in the prevalence of risk factors associated with diet and lifestyle [4]. Obesity and physical inactivity are two factors strongly associated with colorectal cancer [38] and are increasing nationally. Currently, 17% of the population is physically inactive and over 70% are overweight or obese [39]. These factors are modifiable, indicating potential for prevention. Given that this this cancer has one of the highest incidence and mortality rates, it should be a priority in the country’s public policies design. This should focus on improving access to screening tests and early detection, which would help to reduce the number of cases and deaths.

Prostate cancer is one of the most significant tumors of the genitourinary system and has the second-highest incidence rate. The incidence of this ailment is increasing. However, Mexico has a relatively low incidence rate compared to other Latin American and Caribbean countries, ranking higher only than Argentina, Honduras, Ecuador, and Bolivia [40]. Testicular cancer is another type of cancer that affects the genitourinary system and is relevant in Mexico. The national incidence of this cancer differs from that observed in the rest of the world. However, there is a lack of studies on the epidemiology and risk factors of this cancer type in Mexico. Therefore, more research is necessary to implement public policies that could reduce the burden of this disease. Although the global incidence rate has increased, the mortality rate has improved. Unfortunately, Mexico has one of the highest mortality rates [41].

Breast and cervical cancer are the most frequently diagnosed gynecological cancers. Breast cancer has consistently ranked as the third most prevalent malignant neoplasm over the past three decades. In 2019, it represented 7.67% of total deaths, ranking fifth, with a slight increase over the period. The increase in obesity prevalence in Mexico can be attributed to various factors, including changes in diet, physical activity, reproductive choices, and detection at a more advanced stage [42,43]. Studies have found that obesity is a significant risk factor for developing this neoplasia regardless of socioeconomic level, region, or locality [44]. Breast cancer disproportionately affects lower-income populations, but it can impact women of all ages and income levels [12]. In 2002, the Ministry of Health extended and expanded official health regulations and legislation concerning the management of breast cancer through an official technical directive. Mexico has seen a significant decrease in cervical cancer incidence in recent years due to successful campaigns aimed at combating human papillomavirus (HPV). This is because most cervical cancer cases are caused by chronic infection by oncogenic subtypes of HPV. In 2012, Mexico launched a vaccination program for girls aged 11 years to reduce the overall burden of cervical cancer and other diseases caused by HPV [45]. Around 80% of 11-year-old girls in Mexico were covered by vaccination programs. However, vaccination coverage has drastically decreased in recent years, from 11.22% in 2020 to 0.45% in 2021. This decline is mainly due to health efforts being focused on managing the COVID-19 pandemic [11].

Reducing cancer in Mexico is crucial due to disparities in the public healthcare system. The system is primarily composed of employment-based social security systems, and medical services for oncological diagnosis and prognosis are inadequate due to limited infrastructure. Mexico has only 212 radiotherapy units, 358 nuclear medicine units, and 2582 imaging units for the entire population [46]. Therefore, the Mexican government should allocate more resources to improve medical coverage. To achieve more accurate rates, new diagnostic tools and improved registration systems are necessary. National and local governments require additional resources for healthcare and epidemiological surveillance of cancer. Additionally, medical public campaigns for primary healthcare are lacking to increase awareness of the risk factors associated with cancer groups that affect the Mexican population and to prevent an increase in preventable cancer cases. The accurate diagnosis and registration of all cancer cases, including oncological characteristics such as cancer type, staging, and follow-up, are necessary for reliable data. It is important to note that people from rural communities are often excluded from these statistical surveys due to the lack of universal healthcare. The Mexican Congress has established a national cancer registry. However, since its establishment in 2017, it has only included the registration of childhood cancer cases. Therefore, it needs to be expanded to cover the entire population.

This observational study is based on a secondary analysis of data generated by the GBD for Mexico. Therefore, some of the estimates presented here should be interpreted with caution due to the following limitations. The challenges in cancer data collection include (1) insufficient cancer-specific data categorized by year, age, and sex; (2) limited availability of diverse data sources, such as nationally representative studies; (3) varying levels of completeness and time lags in high-quality data; (4) a reduced number of studies on cancer risk factors; and (5) high variability in the data, which may lead to an underestimation of the number of people affected by cancer.

## 5. Conclusions

Cancer remains a significant public health issue in Mexico, marked by rising incidence and mortality rates. While these statistical trends do not directly apply to an individual patient, they play a crucial role in aiding governments, policymakers, health professionals, and researchers in comprehending the impact of cancer on the population. These trends enable them to develop strategies to reduce the cancer burden and to manage and treat cancer. The increasing burden of cancer in Mexico emphasizes the necessity for sustained efforts to address underlying risk factors and enhance access to screening, treatment, and patient care. With appropriate strategies and investments, it is feasible to mitigate the impact of cancer on the health of the Mexican population and improve outcomes.

## Figures and Tables

**Figure 1 cancers-16-00149-f001:**
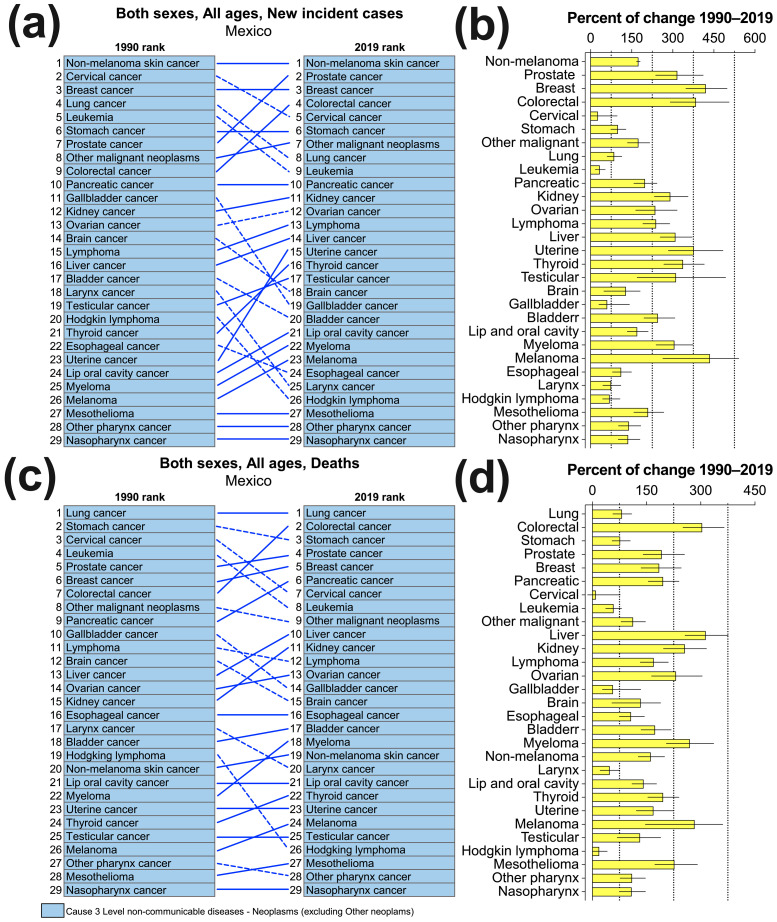
National ranking with percentage of annual change from 1990 to 2019 in the number of cases (**a**,**b**) and deaths (**c**,**d**) by cancer group.

**Figure 2 cancers-16-00149-f002:**
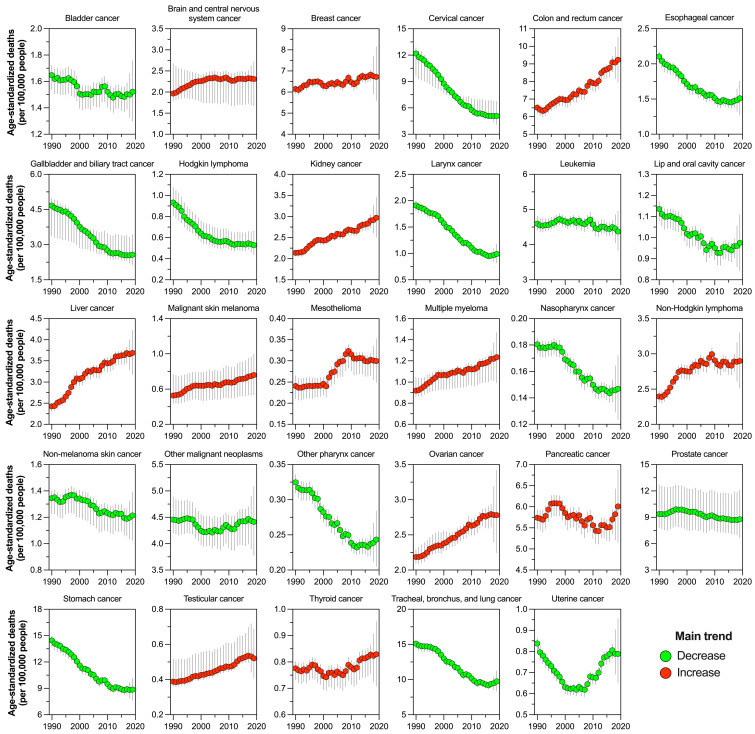
Cancer-specific trends from 1990 to 2019 of the age-standardized death rates (per 100,000) in Mexico.

**Figure 3 cancers-16-00149-f003:**
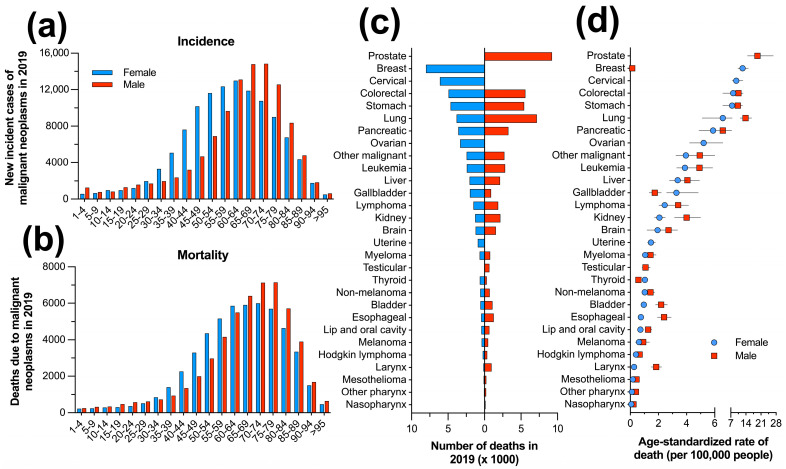
Sex- and age-specific distribution of the incident cases (**a**) and deaths (**b**) due to malignant neoplasms in Mexico during 2019, cancer-specific death counts (**c**), and age-standardized rate of death per 100,000 people by sex (**d**).

**Figure 4 cancers-16-00149-f004:**
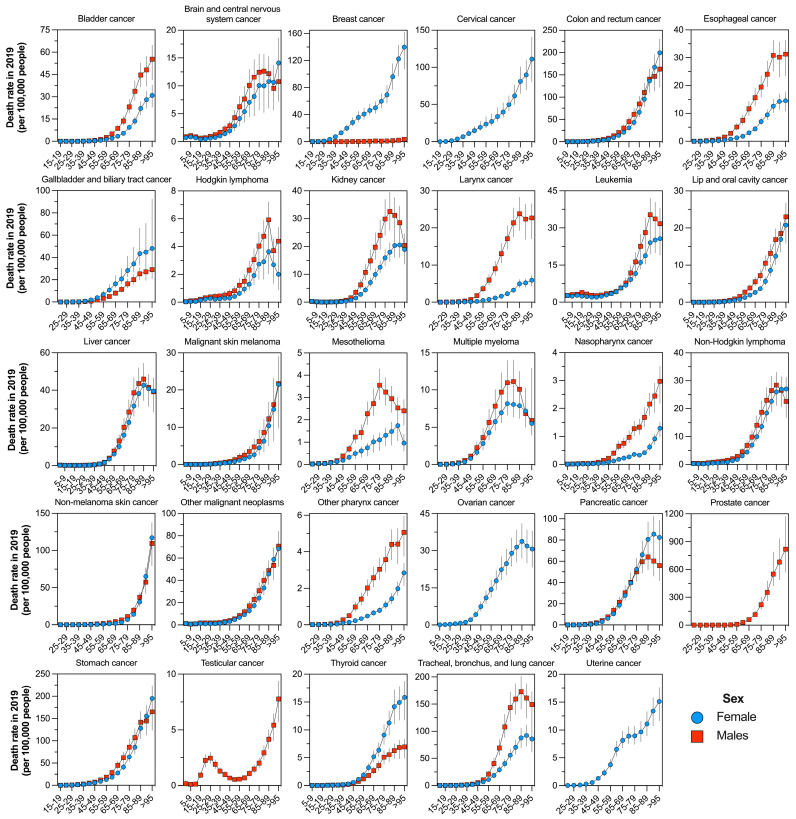
Age-specific rate of death (per 100,000 people) by sex for each group of cancers in Mexico during 2019.

**Table 1 cancers-16-00149-t001:** Total and sex-specific incidence and mortality by cancer group in Mexico in 2019.

Group of Cancer	Incidence (Counts, 95% UI)			Deaths (Counts, 95% UI)		
	Total	Female	Male	Total	Female	Male
Bladder cancer	3071 (2633 to 3597)	1009 (835 to 1224)	2062 (1672 to 2529)	1665 (1417 to 1929)	571 (473 to 689)	1094 (896 to 1321)
Brain and central nervous system cancer	3473 (2515 to 4098)	1547 (958 to 1954)	1925 (1355 to 2369)	2777 (2034 to 3274)	1219 (754 to 1540)	1558 (1099 to 1928)
Breast cancer	24,442 (19,918 to 29,949)	24,312 (19,777 to 29,810)	130 (104 to 161)	8097 (6718 to 9852)	8024 (6649 to 9777)	73 (60 to 90)
Cervical cancer	12,195 (9656 to 16,527)	12,195 (9656 to 16,527)	-	6104 (4904 to 8120)	6104 (4904 to 8120)	-
Colon and rectum cancer	17,470 (15,042 to 20,060)	8044 (6598 to 9771)	9426 (7750 to 11,611)	10,518 (9036 to 12,022)	4917 (4101 to 5914)	5601 (4599 to 6802)
Esophageal cancer	1668 (1417 to 1962)	443 (367 to 533)	1226 (990 to 1495)	1720 (1456 to 2008)	453 (375 to 544)	1267 (1018 to 1541)
Gallbladder and biliary tract cancer	3109 (2591 to 4180)	2139 (1706 to 3137)	970 (761 to 1226)	2910 (2415 to 3865)	2005 (1588 to 2960)	905 (697 to 1150)
Hodgkin lymphoma	1383 (1147 to 1769)	572 (431 to 862)	811 (639 to 1053)	632 (520 to 806)	252 (186 to 366)	381 (301 to 483)
Kidney cancer	6321 (5374 to 7371)	2451 (2032 to 2966)	3870 (3069 to 4794)	3461 (2918 to 4045)	1278 (1057 to 1541)	2182 (1704 to 2730)
Larynx cancer	1493 (1241 to 1789)	216 (174 to 297)	1277 (1033 to 1562)	1118 (926 to 1333)	162 (131 to 223)	957 (775 to 1159)
Leukemia	7787 (6810 to 8825)	3677 (3103 to 4338)	4111 (3469 to 4853)	5249 (4603 to 5958)	2431 (2049 to 2867)	2818 (2351 to 3372)
Lip and oral cavity cancer	1913 (1648 to 2195)	793 (654 to 955)	1121 (908 to 1366)	1109 (958 to 1264)	439 (366 to 529)	670 (552 to 813)
Liver cancer	3973 (3436 to 4550)	1943 (1611 to 2345)	2030 (1661 to 2472)	4183 (3606 to 4792)	2054 (1705 to 2475)	2128 (1731 to 2586)
Malignant skin melanoma	1725 (1296 to 2243)	846 (516 to 1111)	879 (543 to 1281)	874 (618 to 1158)	382 (224 to 500)	492 (305 to 736)
Mesothelioma	439 (364 to 517)	146 (87 to 186)	294 (237 to 362)	353 (294 to 415)	115 (69 to 146)	238 (193 to 289)
Multiple myeloma	1873 (1512 to 2206)	892 (692 to 1100)	981 (687 to 1252)	1444 (1152 to 1714)	664 (529 to 820)	780 (550 to 1001)
Nasopharynx cancer	213 (181 to 251)	62 (51 to 75)	151 (121 to 188)	174 (146 to 204)	45 (37 to 55)	129 (103 to 158)
Non-Hodgkin lymphoma	5069 (4363 to 5862)	2202 (1809 to 2702)	2867 (2335 to 3487)	3399 (2944 to 3876)	1521 (1262 to 1841)	1878 (1526 to 2290)
Non-melanoma skin cancer	41,320 (34,279 to 48,728)	21,263 (17,507 to 25,200)	20,057 (16,717 to 23,577)	1286 (1093 to 1478)	583 (482 to 695)	703 (567 to 851)
Other malignant neoplasms	10,938 (9495 to 12,555)	5625 (4695 to 6724)	5313 (4325 to 6389)	5161 (4415 to 5949)	2437 (1997 to 2937)	2724 (2195 to 3313)
Other pharynx cancer	332 (280 to 392)	89 (73 to 107)	243 (194 to 300)	280 (235 to 327)	68 (57 to 82)	212 (168 to 258)
Ovarian cancer	5341 (4256 to 6599)	5341 (4256 to 6599)	-	3340 (2688 to 4120)	3340 (2688 to 4120)	-
Pancreatic cancer	6674 (5744 to 7683)	3463 (2848 to 4191)	3212 (2579 to 3941)	6853 (5866 to 7853)	3583 (2973 to 4308)	3270 (2649 to 3966)
Prostate cancer	27,097 (20,602 to 36,017)	-	27,097 (20,602 to 36,017)	9256 (7077 to 12,679)	-	9256 (7077 to 12,679)
Stomach cancer	11,272 (9786 to 13,049)	5298 (4393 to 6401)	5973 (4926 to 7343)	10,095 (8688 to 11,605)	4656 (3860 to 5612	5439 (4465 to 6589)
Testicular cancer	3495 (2050 to 4892)	-	3495 (2050 to 4892)	662 (520 to 854)	-	662 (520 to 854)
Thyroid cancer	3534 (2967 to 4174)	2589 (2094 to 3163)	945 (755 to 1168)	944 (811 to 1089)	636 (521 to 768)	309 (244 to 381)
Tracheal, bronchus, and lung cancer	10,890 (9400 to 12,582)	3961 (3271 to 4801)	6930 (5682 to 8476)	11,002 (9425 to 12,698)	3827 (3121 to 4659)	7176 (5881 to 8671)
Uterine cancer	3550 (2870 to 4333)	3550 (2870 to 4333)	-	923 (760 to 1121)	923 (760 to 1121)	-

## Data Availability

All data used are available at https://ghdx.healthdata.org (accessed on: 31 October 2023).

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
