# Peer review of "National Burden and Trends for 29 Groups of Cancer in Mexico from 1990 to 2019: A Secondary Analysis of the Global Burden of Disease Study 2019"

_cancers, 2023, doi:10.3390/cancers16010149_

Round 1

Reviewer 1 Report

Comments and Suggestions for Authors

In their valuable study, Saul A. Beltran-Ontiveros and coworkers describe incidence, mortality, and disability-adjusted lived years due to 29 groups of cancer (by sex and age) as registered in Mexico from 1990 to 2019.

Using the public data from the Global Burden of Disease Study 2019, the authors assessed the national burden of cancer using counts and crude and age-standardized rates per 100,000 people, with 95% uncertainty intervals for 2019 and their trends using the annual percentage of change from 1990 to 2019.

The results obtained provide an exhaustive panorama of oncological epidemiology in Mexico.

The data presentation is more than satisfactory. Some shortening in the Result and Discussion sections would be welcome.
The English editing is satisfactory.  

The authors should make an additional effort to shorten the manuscript, possibly moving some "minor" relevant information among the (already provided) supplementary material.

Interestingly, the manuscript focuses on cancer trends from 1990 to 2013. Such an effort does represent an adjunctive value of the study, also introducing non-minor criticisms about the consistency of the achieved results.
Considering the long-lasting registration time (1990-2013), my personal experience in dealing with cancer registration and epidemiological trends suggests that several registration weaknesses can bias data consistency, which may ultimately affect the obtained results.
This issue is so crucial that a paragraph should be introduced in the main text (not within supplementary data), alerting the readers about some unavoidable weaknesses linked to cancer registration (incidence and death). This is particularly relevant in large countries that include regions with unequal efficiency of health care systems, unequal income levels, and unequal environmental and risk factors). This specific issue should be addressed within the Discussion.

Few comments about the Discussion. The Discussion is well organized but exceedingly long, and several of the provided considerations are well-known. I would suggest shortening: this strategy will not lower the attractiveness of the study.

Comments on the Quality of English Language

Already done within the comments

Reviewer 2 Report

Comments and Suggestions for Authors

The increase in cancer is a verifiable fact; the authors comparatively describe the incidence, mortality and disability-adjusted life years (DALYs) due to 29 cancer groups in Mexico from 1990 to 2019 by sex and age.

The study is extensive and well carried out, it is really an interesting study that provides a lot of information so that researchers can generate new studies on cancer, since the incidence and consequences increase. It is necessary to hypothesize the cause.

As for the presentation, it is difficult to read because there is a lot of information, perhaps Table 2 can be segmented into several tables, since the discussion is partial and is understood much better.

It is the only difficulty that I see with the article, that it is difficult to read, therefore, it would be appreciated to partialize the information a little to be able to read it better.

Reviewer 3 Report

Comments and Suggestions for Authors

The study aims to comparatively describe by sex and age the incidence, mortality, and DALYs due to 29 groups of cancer in Mexico in 2019 with trends from 1990 to 2019. This observational study represents a secondary analysis of the results published in the Global Burden of Disease Study 2019 (GBD 2019). This study was produced and supervised by members of the GBD Collaborative Network from Mexico and complied with the protocol of the GBD.

The Authors described the patterns of cancer incidence among several demographic cohorts, encompassing aspects such as diagnosis rates, mortality rates, mean age of diagnosis, and overall disease burden.

The Authors concluded that the rising cancer burden in Mexico highlights the necessity for continuing efforts to address the underlying risk factors and improve access to screening tests, treatment, and patient care.

It is a quite interesting and very important paper. This is an observational study.

the strength of this paper: very interesting topic, introduction-relevant and concise; the material and methods-the right choice of methodology methods, which was presented in a comprehensible way; the obtained results are presented in the form of figures and tables, which are clear and easy to understand; the discussion- supports the results properly and properly refers to the current literature; the conclusions- based on the obtained results.

There are some comments in the reviewer's opinion which should be taken under consideration by the Authors:

1. In the discussion section, please add the limitations of your study

2.  please add the section -future  strategies to decrease cancer burden in Mexico.
